# Federated DRL-based Coordination of Multi-UAVs for Wildfire Tracking

## Abstract

Formation control continues to pose significant challenges in the field of multi-agent deep reinforcement learning (DRL). This paper presents a formation strategy for multiple UAVs engaged in large-scale wildfire tracking. The proposed approach leverages the Deep Deterministic Policy Gradient (DDPG) algorithm to enable individual UAVs to adapt their path planning and control policies in real time. Although effective for single-UAV scenarios, standard DDPG does not scale well to multi-UAV coordination. Moreover, wildfire fronts rarely evolve symmetrically, as environmental factors such as wind, terrain, and fuel conditions can accelerate fire spread in certain directions, producing irregular boundaries that complicate the maintenance of uniformly spaced formations. To address these limitations, the proposed framework integrates Federated Learning (FL) with DDPG to facilitate collaborative policy refinement without exchanging raw data, introducing a novel performance-weighted federated averaging scheme that prioritizes policies from UAVs demonstrating better formation stability. While conventional FL uses equal-weight aggregation, our approach applies distance- and performance-based weighting to better handle the non-IID data distributions that arise from asymmetric wildfire fronts. These asymmetries—driven by factors such as wind direction and terrain—produce irregular fire boundaries that disrupt uniform formations, making weighted aggregation essential for stable coordination. We apply FL in a novel way to the DDPG components governing linear velocity and its corresponding control gain, both of which are critical for acceleration control and inter-UAV spacing. The simulation results indicate that our method, FL-DDPG, yields significantly improved formation stability—with 2.5 m average spacing variance compared to 14 m for standard DDPG—and improves the average episode reward from –355.45 to –122.21. Overall, the results underscore the importance of performance-weighted aggregation in achieving robust, decentralized coordination in complex wildfire environments.

## 1 Introduction

Wildfires present a serious threat to ecosystems, wildlife, and the economy, driven largely by factors such as climate change and human activity. Uncrewed Aerial Vehicles (UAVs) have become indispensable tools, offering scalable and collaborative capabilities for efficient wildfire monitoring and rapid response across large areas. Advanced path planning and control strategies are essential, with machine learning techniques, particularly reinforcement learning (RL), to enhance the UAV's decision-making (Marshall et al., 2021). Robotic control problems, such as UAV flight, are inherently continuous-time in nature. Recent advancements in continuous-time RL are particularly promising, as they align more naturally with the dynamics of physical systems (Hansen et al., 2023; Wallace & Si, 2025).

Several studies have focused on decentralized control techniques for UAV-based fire tracking. The leader-follower approach, as discussed in (Ghamry & Zhang, 2016), designates one UAV as the leader while others follow predefined relative positions. Although this method simplifies coordination, it suffers from formation instability if the leader fails. To enhance adaptability, behavior-based methods assign UAVs specific behaviors such as dynamic reconfiguration (Feng & Katupitiya, 2022). However, these methods often require an additional adaptation mechanism, which can limit their scalability. Artificial Potential Field (APF) techniques have also been applied to fire track-

ing, leveraging attractive and repulsive forces to preserve formation integrity. Although APF-based control is computationally efficient, it is prone to local minima, which can result in suboptimal trajectories (Seraj & Gombolay, 2020). Virtual structure methods impose rigid formations that enhance stability, but often lead to increased computational overhead and reduced adaptability in dynamic environments (Patrinopoulou et al., 2024).

The Deep Reinforcement Learning (DRL) paradigm provides a framework for agents to learn adaptive policies through environmental interaction. However, in multi-UAV formation control, a fundamental challenge is scalability and coordination. In fully decentralized approaches like independent DDPG, each UAV learns a policy based solely on its local experience. This fails to produce a coherent team-wide strategy, often resulting in poor formation keeping and potential collisions. Consequently, centralized training with decentralized execution algorithms such as Value Decomposition Networks (VDN) have been developed to introduce stronger cooperative incentives by decomposing the global value function during training (Viseras et al., 2021). While VDN can foster cooperation, it is computationally heavy, difficult to scale to large teams, and struggles in asymmetric environments where agent experiences become non-IID. Beyond these algorithmic approaches, recent high-level paradigms have sought to incorporate human-like reasoning into coordination. For instance, (Siedler & Gemp, 2025) use a central Large Language Model (LLM) to mediate strategic interventions, mimicking a human operator. While promising for high-level guidance, such methods introduce a central point of failure and rely on frequent communication, making them less suitable for low-latency, decentralized control in dynamic wildfire scenarios. Federated Learning with Reinforcement Learning (FLRL) offers a promising alternative by enabling agents to collaboratively learn policies without sharing raw data. Recent advancements in FL have shown promising applications in multi-agent configurations, particularly through the FLDDPG approach (Na et al., 2023). In the domain of coordination and control, (Zeng et al., 2022) presents a PID-based control approach in which the controller gains are adaptively updated through a federated learning framework. However, the standard Federated Averaging (FedAvg) algorithm is not well-suited for dynamic control tasks with non-IID data distributions, as it treats all local models equally regardless of their performance. This paper's central contribution is a novel, performance-weighted federated aggregation scheme designed specifically for multi-agent coordination in asymmetric environments.

A comparative overview of existing UAV formation coordination strategies for wildfire tracking is provided in Table 3 (Appendix A), highlighting their key features and limitations.

This study addresses UAV coordination using FLDRL. It builds upon the planning and control framework presented in (Raoufi et al., 2025), which was originally designed for a single UAV. However, as mentioned earlier, achieving a uniform distribution of agents is not straightforward with DDPG with the increase in the number of UAVs. Standard DRL algorithms, such as DDPG, face significant scalability challenges when applied to multi-agent systems due to communication bottlenecks and the physical distribution of the agents (Lan et al., 2025). To address this, we develop a Federated Learning-based DDPG (FLDDPG) framework to enable global coordination across multiple UAVs. Additionally, since standard averaging methods in FL often fall short in dynamic environments, we incorporate additional performance metrics and introduce reward-based feedback mechanisms. These adaptations allow the system to dynamically adjust and improve cooperative strategies in response to changing conditions.

Unlike prior methods that require a central server to maintain a complex joint critic and perform computationally intensive backpropagation—such as the VDN approach used for wildfire tracking in (Viseras et al., 2021)—our FLDDPG framework simplifies the server's role to lightweight model aggregation. Furthermore, while prior works like (Julian & Kochenderfer, 2019) train a single policy offline with no explicit mechanism for organized perimeter coverage, our federated design enables distributed, on-device learning and online adaptation. This makes FLDDPG more practical for large-scale cooperative wildfire monitoring, where operational requirements such as real-time coordination and maximum fire-front coverage are critical. This approach advances beyond foundational FLDDPG implementations for generic tasks, such as (Na et al., 2023), by introducing a performance-weighted aggregation scheme specifically engineered for dynamic, asymmetric environments like wildfire propagation. Where standard federated averaging treats all agents equally, our method ensures that the global model is disproportionately influenced by UAVs that successfully maintain the desired formation.

The remainder of this paper is organized as follows. Section 2 presents the preliminaries, including the multi-UAV motion model, fire front representation, a review of the DDPG algorithm, and the fundamentals of the FL paradigm. Section 3 details the proposed FLDDPG-based formation coordination framework, including the performance-weighted federated aggregation strategy. Section 4 describes the simulation setup and performance evaluation metrics. Finally, Section 5 concludes the paper and outlines future research directions.

## 2 BACKGROUND

This section introduces the mathematical foundations of the proposed framework, including the multi-UAV motion model, the fire-front representation, the DDPG algorithm, and the FL paradigm.

### 2.1 MULTI-UAV MODEL

The motion of a fleet of fixed-wing UAVs is governed by the following 2D equations of motion. Each UAV within the fleet is described by the system of differential equations:

$$
\begin{cases}
\dot{x}_i = v_i(t)\cos(\phi_i(t)) \\
\dot{y}_i = v_i(t)\sin(\phi_i(t)) \\
\dot{\phi}_i = \omega_i(t) \\
\dot{v}_i(t) = a_i(t)
\end{cases}
\tag{1}
$$

In this equation, $x_i$ and $y_i$ represent the position of UAV $i$ in Cartesian coordinate system. The variable $v_i(t)$ denotes the speed of UAV $i$. The orientation angle of each UAV is denoted by $\phi_i(t)$, with $\omega_i(t)$ representing its angular velocity and $a_i(t)$ its linear acceleration.

The following constraints are imposed: $|v_i(t)| \leq v_{sat}, \quad |a_i(t)| \leq a_{sat}, \quad |\dot{\phi}_i(t)| \leq \dot{\phi}_{sat}$ where $v_{sat}, a_{sat}$, and $\dot{\phi}_{sat}$ are the saturation limits for velocity, acceleration, and rate of change of heading angle, respectively.

### 2.2 FIRE FRONT

The fire front is implicitly represented as the zero level set of a scalar function, denoted $\Phi(x, y, t)$. Mathematically, $\Phi(x, y, t)$ is defined as:

$$
\left\{ \Phi(x, y, t) : \mathbb{R}^2 \times \mathbb{R}^+ \to \mathbb{R} \mid \Phi(x, y, t) = 0 \right\},
$$

where $(x, y)$ represents spatial coordinates in a 2D plane, $t$ is time, and $\Phi(x, y, t)$ is a signed distance function that implicitly defines the fire front boundary at time $t$. Based on this definition, the fire front location is where this function is zero, burned area is the places where this function is negative, and the unburned area is where this function is positive.

Computing $\Phi_x = \frac{\partial \Phi}{\partial x}$, $\Phi_y = \frac{\partial \Phi}{\partial y}$, given information about the speed of the fire front. Then, the tangential direction $(T_x, T_y)$ is computed by Equation 2 (Rehm, 2009).

$$
T_x = -\frac{\Phi_y}{\sqrt{\Phi_x^2 + \Phi_y^2}}, \quad T_y = \frac{\Phi_x}{\sqrt{\Phi_x^2 + \Phi_y^2}}.
\tag{2}
$$

The pair $(T_x, T_y)$ corresponds to a clockwise rotation; for consistency across the controller design, we adopt the counterclockwise convention above throughout this work.

The dynamic propagation of the fire front is computed using Equation 3.

$$
\Phi \leftarrow \Phi - dt \cdot U \cdot \sqrt{\Phi_x^2 + \Phi_y^2 + \epsilon}
\tag{3}
$$

where $U$ is the fire spread rate, and the term $\epsilon$ is introduced to prevent singularities.

## 2.3 DDPG ALGORITHM

The DDPG framework utilizes two main components: the actor and the critic networks. The actor network, denoted as $a_t = \mu(s_t \mid \theta^\mu)$, selects the action $a_t$ given the current state $s_t$, while the critic network, denoted as $Q(s_t, a_t \mid \theta^Q)$, evaluates the action chosen by computing its associated Q-value. Here, $\theta^\mu$ and $\theta^Q$ are the parameters of the actor and critic networks, respectively.

During the training process, the agent samples a mini-batch of $N$ transitions, $[s_t, a_t, r_t, s_{t+1}]$, from the experience replay buffer $M$, and updates the critic network by minimizing the loss function.

$$L(\theta^Q) \approx \frac{1}{N} \sum_{k=1}^{N} \left( Y_k - Q(s_k, a_k \mid \theta^Q) \right)^2 \tag{4}$$

where $Y_k$ is the transition target Q-value defined by the Bellman equation: $Y_k = r_k + \gamma Q'(s_{k+1}, \mu'(s_{k+1} \mid \theta^{\mu'}) \mid \theta^{Q'})$ where $\gamma$ is the discount factor, $Q'(s_{k+1}, \mu'(s_{k+1} \mid \theta^{\mu'}) \mid \theta^{Q'})$ represents the predicted Q-value for the next state $s_{k+1}$ under the target networks for the actor and critic, denoted by $\mu'$ and $Q'$, respectively. The loss function is minimized with respect to the parameters of the critic network $\theta^Q$ to improve the precision of the predictions of the Q-value. In the actor network, the gradient of the objective function with respect to the actor's parameters is computed as:

$$\nabla_{\theta^\mu} J \approx \frac{1}{N} \sum_{t=1}^{N} \nabla_{a_t} Q(s_t, a_t \mid \theta^Q) \nabla_{\theta^\mu} \mu(s_t \mid \theta^\mu) \tag{5}$$

where $\nabla_{\theta^\mu} J$ represents the gradient of the objective function with respect to the actor's parameters, and $Q(s_t, a_t \mid \theta^Q)$ is the Q-value for the current state-action pair $(s_t, a_t)$.

The target networks for both the actor and critic are updated using soft updates, ensuring stability during training:

$$\begin{cases} \theta^{\mu'} \leftarrow \tau\theta^\mu + (1-\tau)\theta^{\mu'} \\ \theta^{Q'} \leftarrow \tau\theta^Q + (1-\tau)\theta^{Q'} \end{cases}$$

where $\tau \in [0, 1]$ is a hyperparameter that controls the rate of soft updating.

## 2.4 FEDERATED LEARNING (FL)

FL is a machine learning approach in which a shared global model is trained across multiple decentralized agents that hold local data sets $D_i$. Training is coordinated by a central server that aggregates locally computed updates $w_i$.

$$w_t = \sum_{i=1}^{N} \frac{n_i}{N_t} w_{li},$$

where $w_t$ is the new global model, $w_{li}$ indicates the updated model parameters after local training in the agent $i$. Furthermore, $N$ is the number of agents, $N_t$ where $N_t = \sum_{i=1}^{N} n_i$ is the total number of data points across all agents, and $n_i = |D_i|$: the number of training samples in agent $i$. This formulation corresponds to the standard Federated Averaging (FedAvg) algorithm, which serves as the baseline that we will later extend with performance-aware weighting to better suit dynamic wildfire tracking scenarios.

FL can be categorized into two main types: 1) Horizontal FL (HFL), which is used when agents have the same feature space but different samples. The above relationships are given for HFL. 2) Vertical FL (VFL), which is used when agents have different features but share the same samples.

In this study, we employ HFL, where all UAVs share the same feature space (e.g. GPS position, velocity, and fire front gradients), but collect different samples from their respective local environments. Although the wildfire front evolves asymmetrically due to wind, terrain, and fuel distribution, the learning setup remains HFL because the feature representation across UAVs is identical. The asymmetry primarily results in non-independent and identically distributed (non-IID) data distributions across clients, meaning some UAVs experience faster-spreading boundaries while others observe slower dynamics. To address this, our framework integrates a performance-weighted federated aggregation strategy, ensuring that UAVs contributing more stable formation tracking have

greater influence on the global model. This preserves the HFL structure while improving robustness under asymmetric fire-front evolution.

# 3 MULTI-UAV POLICY AGGREGATION AND COORDINATION

Path planning and control for each UAV are based on the approach presented in (Raoufi et al., 2025). As illustrated in Figure 1, the system implements a two-layer autonomous control architecture for each UAV. The high-level planner uses a DDPG neural network to generate optimal flight commands by processing the UAV's current state and environmental information, including fire properties. It outputs a desired velocity and heading angle $(v_{d_i}, \phi_{d_i})$ to efficiently track the wildfire front while maintaining formation with other UAVs. The low-level controller then translates these planned commands into precise flight adjustments. It continuously compares the desired path with the UAV's actual measured velocity and heading, computes any tracking errors, and applies adaptive control gains to generate the corresponding linear acceleration and angular velocity $(a_{d_i}, \dot{\phi}_{d_i})$. This ensures the UAV accurately follows the planned trajectory despite environmental disturbances. The system operates in a continuous closed loop: the UAV executes the low-level commands, observes new environmental states and fire front data, and feeds this information back to the high-level planner for the next decision cycle, enabling real-time adaptation to the dynamic wildfire environment.

For a fixed-wing UAV operating in a 2D environment, the primary control inputs are the turn rate and throttle change, which directly influence the UAV's heading and speed, respectively.

$$\begin{cases} \dot{\phi}_{\text{uav}_i} = K_{\phi_i} e_{\phi_i}, \\ \dot{v}_{\text{uav}_i} = K_{v_i} e_{v_i} \end{cases} \tag{6}$$

where $K_v$ and $K_\phi$ represent the control gains, $e_{\phi_i} = (\phi_{d_i}(t) - \phi_{\text{uav}_i}(t))$, $e_{v_i}(v_{d_i}(t) - v_{\text{uav}_i}(t))$. The RL state, action, and reward for each UAV are defined as follows: The state $(s_i)$, which includes

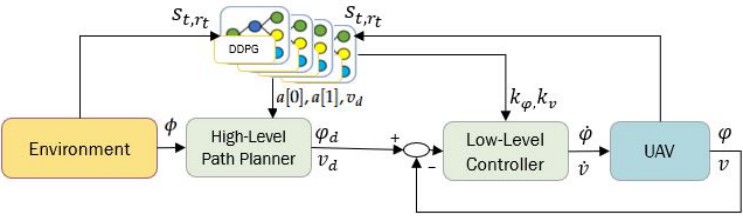

Figure 1: Overview of the DDPG-based planning and control framework for an individual agent (Raoufi et al., 2025).

the UAV's own position $(x_i, y_i)$ and velocity $v_i$, the local fire-front gradient components $(\Phi_x, \Phi_y)$, and the set of Euclidean distances $d_{ij}$ to all other UAVs $j \neq i$ in the fleet.

$$d_{ij} = \sqrt{(x_i - x_j)^2 + (y_i - y_j)^2}$$

$$s_i = \{x_i, y_i, v_i, \Phi_x, \Phi_y, d_{ij}\}$$

$$\mathbf{a_{low_i}} = [K_{v_i}, K_{\phi_i}], \tag{7}$$

$$\mathbf{a_{high_i}} = [a_i[0], a_i[1], v_{d_i}] \tag{8}$$

where $a_{high_i}$ includes the heading correction terms $a_i[0], a_i[1]$. All actions are normalized within the range $[-1, 1]$, except for $K_{\phi_i}, K_{v_i}$, which is constrained to $[0, 1]$.

The planning velocity $v_{d_i}$ is computed directly using DDPG, while the planning heading angle is computed by Equation 9, in which $T_x, T_y$ are direction vectosr and $a_i[0], a_i[1]$ are the DDPG output designed to regulate each UAV heading angle.

$$\phi_{d_i} = \arctan\left(\frac{U.T_y + a_i[1]}{U.T_x + a_i[0]}\right) \tag{9}$$

where $U.\mathrm{T}_x, U.\mathrm{T}_y$ are defined in Equation 2, and $a_i[0]$ and $a_i[1]$ are actions computed by the DDPG algorithm.

The reward is focused on minimizing the distance of each UAV to the fire front, aligning the UAVs' heading with the direction of fire spread, and it encourages the UAVs to maintain proper spacing and coordination within the fleet. Therefore, the reward function of the overall system is defined as

$$r_i = \begin{cases} -\gamma_1 di_{\mathrm{fire}} - \gamma_2 e_{\phi_i, v_i} \\ -\gamma_3 \sum_{j \in \mathcal{N}_i} \left| d_{ij} - d_{\mathrm{form}}^{\mathrm{des}} \right|, & \text{if not in burned area,} \\ -\gamma_1 di_{\mathrm{fire}} - \gamma_2 e_{\phi_i, v_i} \\ -\gamma_3 \sum_{j \in \mathcal{N}_i} \left| d_{ij} - d_{\mathrm{form}}^{\mathrm{des}} \right| - C, & \text{if in burned area,} \end{cases} \tag{10}$$

where $d_{\mathrm{fire},i} = |\Phi(x_i, y_i, t)|$, and $d_{\mathrm{des}}^{\mathrm{form}}$ is the desired spacing between the two neighboring UAVs. In the circular fire-front formation each UAV $i$ monitors only its immediate left and right neighbors. Hence, the neighbor set for each UAV $i$ is defined as its two immediate neighbors, $\mathcal{N}_i = \{i-1, i+1\}$.

The constants $\gamma_1$, $\gamma_2$, and $C$ in Equation 10 are empirically tuned through iterative testing to balance fire-front tracking, heading alignment, and formation spacing. $\gamma_1$ is tuned to prioritize minimizing fire-front distance, $\gamma_2$ to correct heading without overpowering positional objectives, $\gamma_3$ to regulate inter-UAV spacing and encourage stable formation maintenance, and $C$ as a large penalty to discourage entering burned areas.

Building on the single-agent DDPG framework, the multi-UAV problem is handled by allowing each vehicle to train its own actor $\mu_i$ and critic $Q_i$ on local interaction data. For formation control, we hypothesize that linear velocity and its control gain are the primary determinants for regulating inter-UAV spacing. Therefore, we employ a selective federation strategy. Periodically, each UAV transmits only the relevant subsets of its local model parameters—specifically those governing velocity ($\theta_{i,v}$) and its control gain ($\theta_{i,K_v}$)—to a virtual central server (Figure 2). The server computes a weighted aggregation of these parameters; agents that maintain tighter adherence to the desired formation spacing exert greater influence on the global policy, thereby biasing learning toward stability under asymmetric fire-front dynamics. The resulting global parameters are broadcast back to all clients, and local models are updated via a soft-update with rate $\tau_g$ (see Algorithm 1).

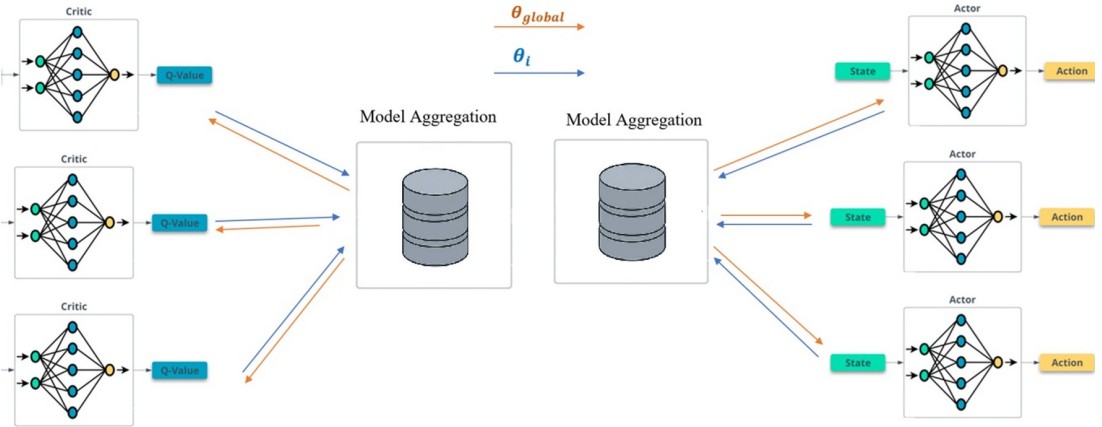

Figure 2: FLDDPG block diagram for three agents.

Figure 2 represents the federated learning framework, where each UAV periodically transmits its local model parameters, including the actor and critic network weights, to a virtual central server. Federated aggregation in this proposed framework does not follow the standard FedAvg method. Instead, it uses the modified weighting rule defined in Equation 12, where the contribution of each UAV is exponentially scaled by its deviation from the desired formation spacing.

The global model parameters are determined by equation 11.

$$
\begin{cases}
\theta_{\text{global},v} = \dfrac{\sum_{i=1}^{N} w_i \cdot \theta_{i,v}}{\sum_{i=1}^{N} w_i}, \\[3mm]
\theta_{\text{global},K_v} = \dfrac{\sum_{i=1}^{N} w_i \cdot \theta_{i,K_v}}{\sum_{i=1}^{N} w_i}
\end{cases}
\tag{11}
$$

where $\theta_i$ denotes the parameter vector for UAV $i$ (coming from the DDPG-updated actor or critic), and the weight $w_i$ is defined as

$$
w_i = \exp\left( -\frac{1}{\sigma} \sum_{j \in N_i} \left| d_{ij} - d_{\text{des}}^{\text{form}} \right| \right),
\tag{12}
$$

where $\sigma$ is a scaling parameter that modulates the sensitivity to formation deviations. This weighted aggregation improves the formation by using feedback from the spacing reward. In essence, UAVs that align more closely with the desired spatial and performance parameters contribute more significantly to the global model.

**Remark** on Motivations and Theoretical Justification for Formation-Weighted Federated Aggregation:

The proposed federated aggregation scheme is motivated by several mechanisms that can improve multi-agent learning. First, due to the asymmetric fire spread, each UAV experiences a distinct local environment. Federated aggregation enables the dissemination of useful strategies learned in these diverse conditions, enhancing the model's ability to generalize to novel scenarios. Second, parameter aggregation acts as a form of implicit regularization, reducing variance and discouraging overfitting to local noise. Third, agents collecting experience in parallel expose the shared policy to a broader set of transitions than any single agent could, accelerating learning in terms of sample efficiency.

## 4 SIMULATION RESULTS

The wildfire scenario in this study is modeled after the Black Friday bushfire that occurred in Victoria, Australia (Cruz et al., 2012). To create a realistic and dynamic wildfire scenario, the fire front evolution generated by FARSITE (Pham et al., 2017) was integrated into our UAV simulation framework. The parameters used in the FARSITE model, along with the terrain data, were calibrated based on historical data from the Kilmore East event, as summarized in Table 1.

Table 1: The Kilmore East wildfire used to calibrate the simulation.

| Parameter | Value |
| --- | --- |
| Main Fuel Type | Grassland/Open Woodland; canopy cover <10% |
| Avg. Spread Rate ($R$) | 1.33 m/s (mixed fuels) |
| Open Wind Speed ($U_0$) | 12.78–15.28 m/s; gusts up to 20.0 m/s |
| Wind Adjustment Factor | 0.4–0.6 (grassland/open woodland) |
| Mid-Flame Wind Speed ($U$) | $U = U_0 \times$ WAF = 5.1–9.2 m/s |

The DDPG agents were trained using a discount factor $\gamma = 0.99$, soft update parameter $\tau = 0.01$, and learning rates of $3 \times 10^{-4}$ for both actor and critic networks. The replay buffer size was set to $10^5$ with a batch size of 64. Federated weight aggregation was applied with $\sigma = 0.5$ and soft update parameter $\tau_g = 0.075$. Regarding the reward setting, UAVs received a penalty with a weight of $\gamma_1 = 20$ per unit distance to the fire front, a heading and velocity error penalty with a constant weight of $\gamma_2 = 20$ (applied in the low-level reward), and a large penalty of $C = 100$ upon entering the burned area. A formation spacing penalty with a weight of $\gamma_3 = 20$ was applied. The

DDPG implementation utilizes Ornstein-Uhlenbeck noise for exploration, which introduces inherent stochasticity during training.

In the first scenario, online path planning and fire front tracking are performed by three fixed-wing UAVs over a large-scale area, with the entire process completed within 10 seconds. The results are presented in Figure 3(left)- 4(left). Figure 3(left) demonstrates that although the UAVs reach the fire front at slightly different times and experience varying environmental conditions—due to the wildfire's asymmetric expansion—they successfully arrange themselves into a coordinated formation using the FLDDPG approach. In this experiment, the wildfire spread asymmetry is modeled by assuming a wind angle of $\pi/4$, which causes the fire to expand more rapidly toward the right and upper regions of the domain. It should be noted that the wind angle information in the Victoria dataset is not explicitly provided; hence, this assumption is introduced for the purpose of evaluation. UAV1, UAV2, and UAV3 are depicted in blue, black, and cyan, respectively. Figure 4(left) illustrates the normalized cumulative FLDDPG rewards for planning and control of each UAV, while the total system reward (normalized) is shown as a solid black line. These results demonstrate that the FLDDPG method facilitates effective collaborative coverage, a well-arranged formation, and efficient reward accumulation across all agents.

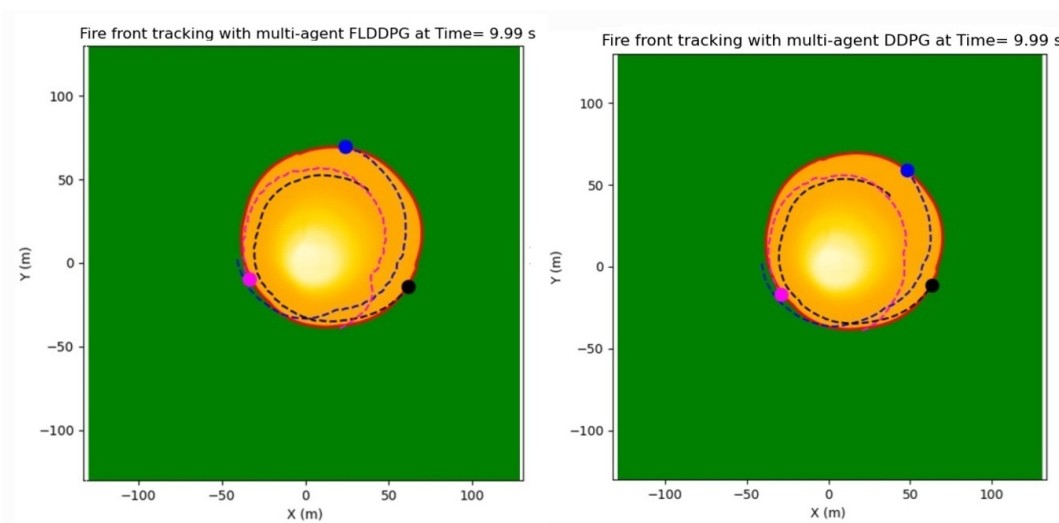

Figure 3: Fire front tracking with three UAVs using FLDDPG-Based RL(left), and DDPG-Based RL(right).

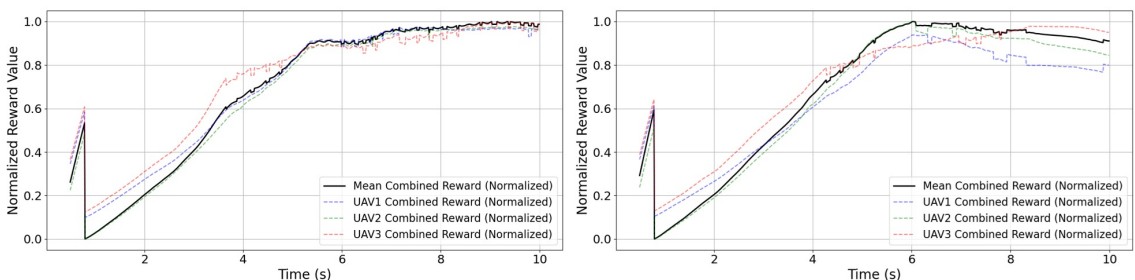

Figure 4: Normalized cumulative total reward progression for individual UAVs and the overall team reward in three-UAV formation using FLDDPG (left) and DDPG (right).

To evaluate the proposed FLDDPG-based multi-agent reinforcement learning (MARL) approach, the simulation was repeated using a standard DDPG-based MARL method for comparison. The UAVs were initialized and configured identically in both scenarios. The results are shown in Fig-

ure. 3(right)- 4(right). At the final time step, the UAVs fail to maintain an organized formation; Notably, the distance between UAV1 (blue) and UAV2 (black) is significantly less than the desired spacing. Furthermore, Figure 4(right) reveals that the total normalized reward curves for each UAV do not exhibit a consistent increasing trend during fire tracking. This outcome is attributed to the erratic progression of the rewards associated with the formation spacing penalty, specifically the term $\gamma_3 \sum_{j \in \mathcal{N}_i} \left| d_{ij} - d_{\text{form}}^{\text{des}} \right|$. Consequently, the overall system reward (solid black line) does not show a steady increase, underscoring the limitations of standard DDPG in multi-UAV coordination tasks.

Table 2: Comparison between DDPG and FLDDPG

| Metric | DDPG | FLDDPG |
|---|---|---|
| Distance $d_{12}$ (m) | 75.9 | 90.69 |
| Distance $d_{13}$ (m) | 111.02 | 96.60 |
| Distance $d_{23}$ (m) | 96.62 | 94.6 |
| Overall Average Reward | -355.45 | -122.21 |

As observed from the comparison between Figure 4 (left) and Figure 4 (right), the FL-DDPG framework introduces a small computational overhead from federated aggregation, which can slow initial policy development compared to independent DDPG learners. Although DDPG may exploit short-term rewards more quickly, this comes at the cost of long-term stability.

For a more comprehensive analysis, the comparative results are presented in Table 2, which clearly demonstrates that the FLDDPG framework generally outperforms the standard DDPG approach in both formation control and cumulative reward attainment. In terms of formation accuracy, FLDDPG achieves a more uniform arrangement, as evidenced by the inter-UAV distances $d_{12}$, $d_{13}$, and $d_{23}$ being closely aligned with each other. In contrast, the DDPG method results in inconsistent and suboptimal spacing, leading to a disorganized formation. This quantitative difference is captured by the standard deviation of the distances: analysis reveals that while both methods achieve a similar target scale (mean distance $\approx 94\,\text{m}$), their structural integrity differs drastically. The conventional DDPG approach yields an inconsistent formation ($\approx 14\,\text{m}$), whereas the proposed FLDDPG method maintains a regular, equilateral geometry ($\approx 2.5\,\text{m}$). The superior performance of FLDDPG is conclusively demonstrated by the overall average reward, a key performance indicator. The FLDDPG framework's reward of -122.21 is greater than that of the DDPG baseline -355.45, highlighting its significantly enhanced capability for coordinated wildfire tracking.

To evaluate the scalability of our proposed framework, we extended the simulation to a more complex scenario involving five UAVs. The UAVs are assigned the following colors for visualization: UAV1 (blue), UAV2 (black), UAV3 (magenta), UAV4 (cyan), and UAV5 (brown). The fire front tracking paths for the five-UAV scenario are visualized in Figure 5. The results for the FL-DDPG framework (left) demonstrate its capability to maintain an organized formation despite the highly asymmetric and dynamic evolution of the wildfire front. In contrast, the standard DDPG approach (right) fails to achieve a coordinated formation. While the DDPG-based UAVs initially approach the fire perimeter, they are unable to establish or maintain a structured spatial configuration, leading to disorganized and inefficient coverage after approximately 20 seconds.

The progression of the total normalized cumulative rewards for each individual UAV and the overall team reward is presented in Figure 6. The FL-DDPG results (left) show a consistent and coordinated increase in both individual and collective rewards. This synchronized upward trend indicates that all agents are successfully contributing to the shared objectives of fire-front tracking and formation maintenance. Conversely, the reward curves for the DDPG method (right) are uncoordinated.

## 5 CONCLUSION

This paper presented a federated deep reinforcement learning framework for multi-UAV formation control in dynamic wildfire tracking, enabling decentralized execution while leveraging centralized collaborative learning. By integrating performance-weighted federated averaging into the multi-agent DDPG structure, the proposed approach enables UAVs to collaboratively refine their policies without sharing raw data, thereby reducing communication overhead. The reward-based weighting

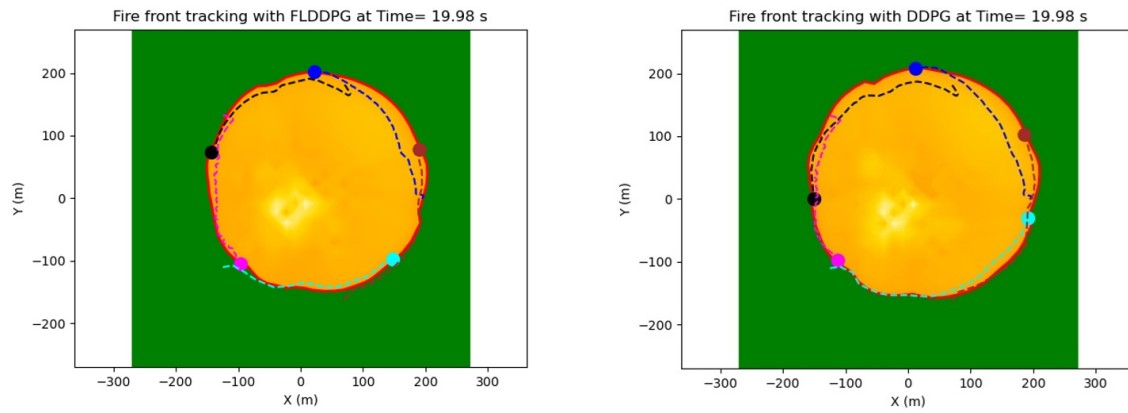

Figure 5: Fire front tracking with five UAVs using FLDDPG-Based RL(left), and DDPG-Based RL(right).

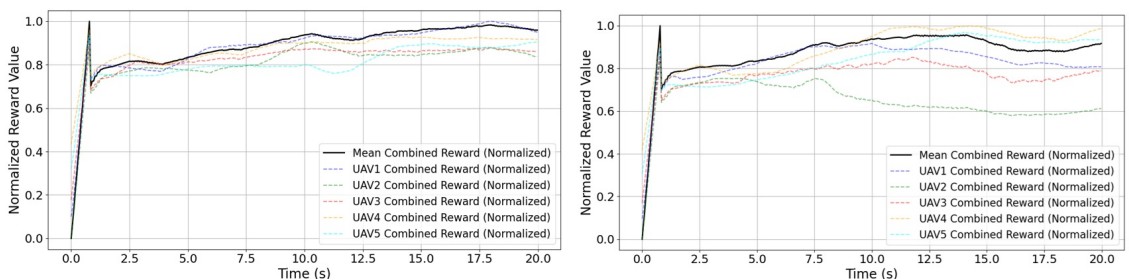

Figure 6: Normalized cumulative total reward progression for individual UAVs and the overall team reward in five-UAV formation using FLDDPG (left) and DDPG (right).

technique ensures that UAVs contributing to improved formation structure and tracking performance have a greater influence on the global model. Simulation results demonstrated that FLDDPG outperforms DDPG in terms of UAV distances and overall average reward, highlighting its effectiveness for real-time, large-scale coordination in evolving fire-front environments.

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

# A SUMMARY OF UAV COORDINATION METHODS IN WILDFIRE MONITORING

Table 3: Overview of UAV formation coordination strategies in wildfire tracking.

| Method | Definition | Advantages | Disadvantages |
|---|---|---|---|
| Leader-follower Method (Ghamry & Zhang, 2016) | One agent is designated as the leader, while the remaining agents function as followers. | Simple implementation; scalable to multiple UAVs | Formation design can be challenging if the leader fails |
| Behavior-based Method (Feng & Katupitiya, 2022) | Agents exhibit multiple prescribed behaviors. | Robust to environmental changes; flexible and adaptable formations | Requires complex behavior tuning; can lead to oscillations or instability |
| Artificial Potential Method (Seraj & Gombolay, 2020) | UAVs navigate using attractive and repulsive forces to reach the target while avoiding collisions. | Simple to implement; provides smooth and continuous motion paths | Can lead to local minima; may struggle to converge to the desired formation |
| Virtual Structure Method (Patrinopoulou et al., 2024) | The agents form a rigid structure guided by a virtual leader. | Maintains a stable formation | Being centralized increases time due to computational management |
| Decentralized RL | Agents learn policies by interacting with the environment to maximize cumulative reward. | Adaptive; no need for explicit models | Lack of global coordination; overfits to local conditions |
| VDN (Value Decomposition Networks) (Viseras et al., 2021) | Decomposes the global value function into individual value functions for each agent. | Strong cooperation among agents; faster convergence in cooperative tasks | Computationally heavy; hard to scale to large teams |
| FL + RL | Enables agents to collaboratively learn policies without sharing raw data. Agents train in parallel and periodically merge models. | Reduces communication bandwidth; increases privacy; efficient in dynamic environments | May suffer from non-IID data issues |

# B    DEFAULT NOTATION FOR FLDDPG FRAMEWORK

To ensure clarity and consistency, we summarize the main symbols used throughout the paper.

**UAV States and Dynamics**

| | |
|---|---|
| $(x_i, y_i)$ | Cartesian position of UAV $i$ |
| $v_i$ | Linear velocity of UAV $i$ |
| $\phi_i$ | Heading angle (orientation) of UAV $i$ |
| $\omega_i$ | Angular velocity of UAV $i$ |
| $a_i$ | Linear acceleration of UAV $i$ |
| $d_{ij}$ | Euclidean distance between UAV $i$ and UAV $j$ |
| $d_{\text{des}}^{\text{form}}$ | Desired inter-UAV formation spacing |

**Fire Front Representation**

| | |
|---|---|
| $\Phi(x, y, t)$ | Level-set function representing wildfire front |
| $\nabla\Phi = (\Phi_x, \Phi_y)$ | Gradient of fire-front level-set |
| $(T_x, T_y)$ | Tangential unit vector to the fire front |
| $U$ | Fire spread rate |

**Reinforcement Learning**

| | |
|---|---|
| $s_i$ | State vector of UAV $i$ |
| $a_i$ | Action chosen by UAV $i$ |
| $r_i$ | Reward of UAV $i$ |
| $\mu(s|\theta^\mu)$ | Actor policy network with parameters $\theta^\mu$ |
| $Q(s, a|\theta^Q)$ | Critic value network with parameters $\theta^Q$ |
| $\gamma$ | Discount factor for future rewards |

**Reward Weights**

| | |
|---|---|
| $\gamma_1$ | Weight for distance to fire front |
| $\gamma_2$ | Weight for heading/velocity alignment error |
| $\gamma_3$ | Weight for formation spacing penalty |
| $C$ | Large penalty for entering burned area |

**Federated Learning**

| | |
|---|---|
| $\theta_i$ | Local model parameters of UAV $i$ |
| $\theta_{\text{global}}$ | Aggregated global model parameters |
| $N$ | Number of UAVs (agents) |
| $n_i$ | Local sample size of UAV $i$ |
| $N_t$ | Total number of samples at iteration $t$ |
| $\tau, \tau_g$ | Soft update rates for local and global target networks |

## C  PSEUDOCODE

Algorithm 1 provides the detailed pseudo-code for the proposed FLDDPG framework.

---

**Algorithm 1**

---

1: **Initialize:**
2: Set initial states for each UAV $i \in \{1, \ldots, N\}$: $(x_i, y_i, v_i, \phi_i)$.
3: Load fire front data $\Phi(x, y, t)$ and compute tangential directions $(T_x, T_y)$.
4: Initialize local actor $\mu_i$, critic $Q_i$, target networks $\mu_i'$, $Q_i'$, and replay buffer $M_i$ for each UAV $i$.
5: Initialize global actor $\theta_{\text{global}}^\mu$ and critic $\theta_{\text{global}}^Q$.
6: Set DDPG and FL hyperparameters.
7: **State observation for each UAV $i$:**
$\quad s_i = \{x_i, y_i, v_i, \Phi_x, \Phi_y, d_{ij}, j \neq i\}$
8: **High-Level Section (Policy Execution):**
9: Query the local actor network $\mu_i$ to compute a high-level action vector based on the current state:
$\quad a_i^{\text{high}} = \mu_i(s_i \mid \theta^\mu) + \mathcal{N}_t \,,$
$\quad$ where $a_i^{\text{high}} = [a_i[0], a_i[1], v_i^d, K_{\phi_i}, K_{v_i}]$, and $\mathcal{N}_t$ is exploration noise that decays over time.
10: Compute Desired Heading Angle:
$\quad \phi_i^d = \arctan\left( \frac{U \cdot T_y + a_i[1]}{U \cdot T_x + a_i[0]} \right).$
$\quad$ The pair $(\phi_i^d, v_i^d)$ defines the desired trajectory setpoint.
11: **Low-Level Control:** $\begin{cases} \dot{\phi}_{\text{uav}_i} = K_{\phi_i} e_{\phi_i}, \\ \dot{v}_{\text{uav}_i} = K_{v_i} e_{v_i} \end{cases}$
12: **UAV dynamics update:** $x_i \leftarrow x_i + v_i \cos(\phi_i)\Delta t, \quad y_i \leftarrow y_i + v_i \sin(\phi_i)\Delta t$
13: **Reward computation for UAV $i$:**

$$r_i = \begin{cases} -\gamma_1 di_{\text{fire}} - \gamma_2 e_{\phi_i, v_i} \\ -\gamma_3 \sum_{j \in \mathcal{N}_i} \left| d_{ij} - d_{\text{form}}^{\text{des}} \right|, & \text{if not in burned area,} \\ -\gamma_1 di_{\text{fire}} - \gamma_2 e_{\phi_i, v_i} \\ -\gamma_3 \sum_{j \in \mathcal{N}_i} \left| d_{ij} - d_{\text{form}}^{\text{des}} \right| - C, & \text{if in burned area,} \end{cases}$$

14: **Store experience:** Append $(s_i, a_i, r_i, s_i')$ into $M_i$.
15: **Local learning:**
16: Sample mini-batch from $M_i$
17: Update critic $Q_i$ using Equation 4
18: Update actor $\mu_i$ using Equation 5
19: Soft update targets: $\theta_i^{\mu'}, \theta_i^{Q'}$
20: **Federated aggregation:**
21: Each UAV $i$ sends parameters $\{\theta_i^\mu, \theta_i^Q\}$ with weight
$\quad w_i = \exp\left( -\frac{1}{\sigma} \sum_{j \in N_i} |d_{ij} - d_{\text{des}}^{\text{form}}| \right)$
22: The server aggregates the global model parameters for the velocity $v$ and its control gain $K_v$ by computing a weighted average of the local parameters from each client,
$\quad \theta_{v,\text{global}} = \frac{\sum_{i=1}^N w_i \cdot \theta_{i_v}}{\sum_{i=1}^N w_i}, \theta_{K_v,\text{global}} = \frac{\sum_{i=1}^N w_i \cdot \theta_{i, K_v}}{\sum_{i=1}^N w_i}$
23: Broadcast $\theta_{\text{global}}$ to all UAVs, update local models.
24: **Termination:** Stop when fire front fully tracked, formation lost, or time limit reached.

---

### C.1 ABLATIONS FOR PERFORMANCE-WEIGHTED AGGREGATION

To evaluate the effectiveness of the proposed performance-weighted federated aggregation strategy and to further examine the selective federation hypothesis, we conducted an ablation study comparing three variants standard DDPG baseline, FLDDPG with FedAvg, and FLDDPG with the proposed performance-weighted aggregation. As shown in Fig. 5 (right), the standard DDPG baseline fails

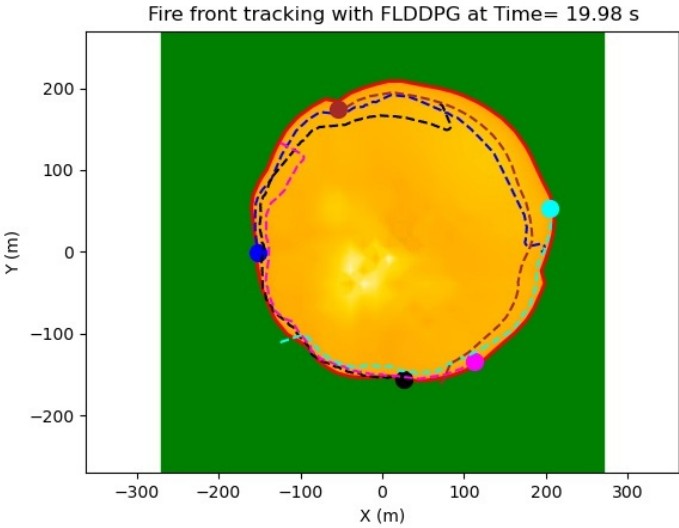

Figure 7: Fire-front tracking by five UAVs using FL-DDPG with a customized federated learning scheme.

to maintain a stable formation, producing disorganized tracking paths and inconsistent inter-UAV spacing. The FLDDPG variant using FedAvg in Fig. 7 improves upon the baseline and achieves better coordination, but still exhibits noticeable formation drift. In contrast, the proposed FLDDPG with performance-weighted aggregation (Fig. 7 (left)) achieves the most stable formation, demonstrating significantly improved coordination behavior and the highest cumulative reward among all tested methods.

Table 4: Average reward comparison of DDPG and FLDDPG variants for 5-UAV wildfire tracking.

| Method | Overall Average Reward |
|---|---|
| Standard DDPG | -871.89 |
| FLDDPG + FedAvg | -520.33 |
| FLDDPG + Weighted (Proposed) | -380.00 |

## C.2 TEMPORAL ANALYSIS OF SWARM BEHAVIOR

To provide deeper insight into the swarm's adaptive response to the dynamic fire front, we analyze the temporal evolution of planning variables. The velocity and heading angle profiles in Fig. 10 and Fig. 9 reveal how each UAV adjusts its behavior in real time to maintain formation and track the evolving fire perimeter. The signals remain within feasible bounds and exhibit correlated adjustments.

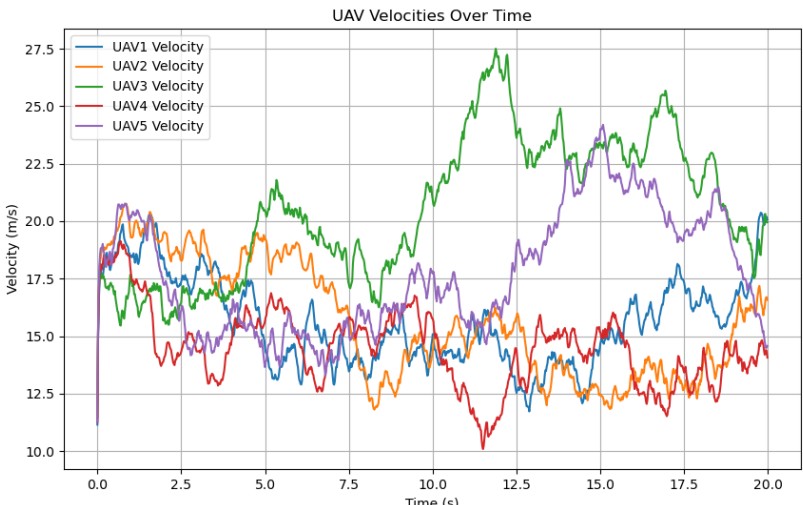

Figure 8: Velocity profiles of each UAV using FL-DDPG, showing coordinated speed adjustments in response to fire front evolution.

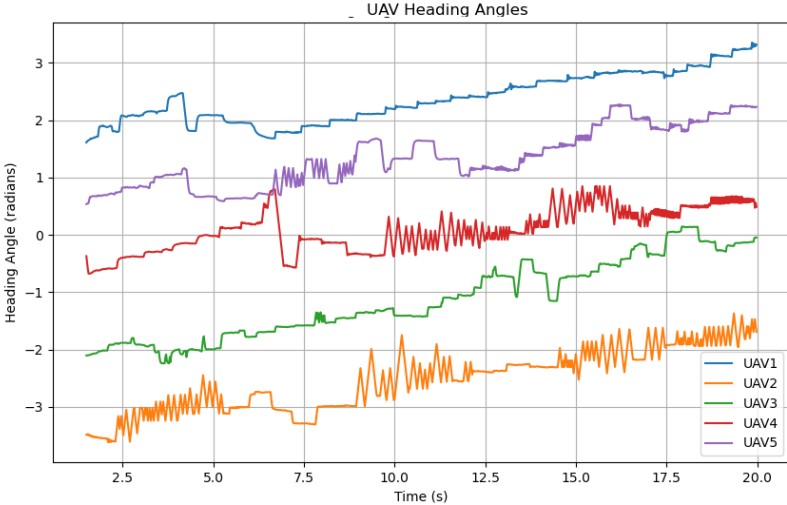

Figure 9: Heading angle evolution for each UAV using FL-DDPG, demonstrating coordinated orientation control for fire front tracking.

### C.3 COMPARISON WITH VALUE DECOMPOSITION NETWORKS

Value Decomposition Networks (VDN) is a class of centralized training with decentralized execution algorithms in cooperative multi-agent reinforcement learning. VDN learns a centralized joint action-value function $Q_{\text{total}}(s, a)$ that is factorized into individual agent utilities $Q_i(s_i, a_i)$, enabling implicit coordination during training while maintaining decentralized execution. To provide a performance comparison, we implemented a VDN (Viseras et al., 2021) baseline using the same multi-UAV wildfire tracking environment. FLDDPG attains an overall average reward of $-380$, which

Table 5: Comparative Performance of VDN and FLDDPG for 5-UAV Wildfire Tracking; The desired spacing for maximum coverage is 243.4.

| Metric | VDN Baseline | Proposed FLDDPG |
|---|---|---|
| Distance $d_{12}$ (m) | 190 | 212 |
| Distance $d_{23}$ (m) | 260 | 220 |
| Distance $d_{34}$ (m) | 173.1 | 237 |
| Distance $d_{45}$ (m) | 236 | 202 |
| Distance $d_{15}$ (m) | 232 | 221 |
| Overall Average Reward | -571.89 | -380 |
| Avg. CPU Usage (%) | 10.0% | 5.2% |

is more than twice as high as VDN's reward of $-571.89$. This substantial improvement directly reflects FLDDPG's superior ability to maintain formation stability, minimize spacing penalties, and achieve consistent fire-front tracking. In terms of computational efficiency, FLDDPG consumes only $5.2\%$ average CPU usage, compared to VDN's $10.0\%$—a reduction of nearly half. This efficiency advantage stems from fundamental architectural differences: VDN incurs high communication and computation costs because it requires transmitting full state–action information or gradients to a central server and maintaining a joint critic whose complexity grows with team size. In contrast, FLDDPG adopts a lightweight federated design that distributes learning across agents and requires only periodic parameter exchanges, drastically reducing both communication overhead and central processing burden. Therefore, FLDDPG not only performs better in terms of task reward and formation accuracy, but does so with significantly lower computational cost and greater scalability, making it more suitable for real-world deployment in dynamic, resource-constrained wildfire environments.

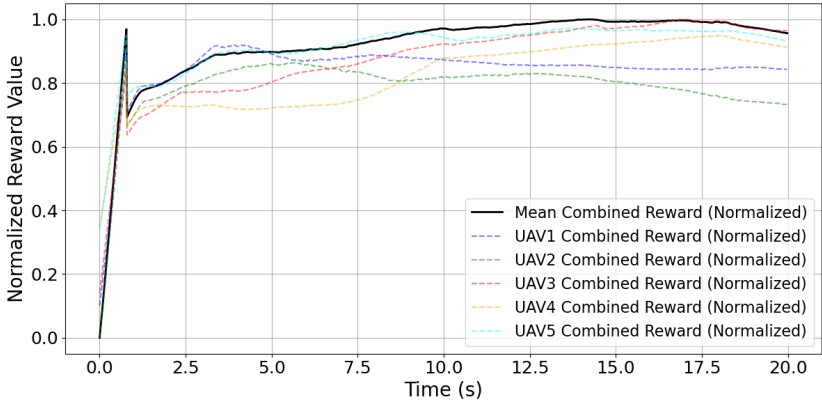

Figure 10: Normalized cumulative total reward progression for individual UAVs and the overall team reward in five-UAV formation using VDN.

