# OpenReview forum: "Federated DRL-based Coordination of Multi-UAVs for Wildfire Tracking"
_ICLR.cc/2026/Conference — Submitted to ICLR 2026_

### Official Review · Reviewer_xqkU · 2025-10-26

**Soundness:** 2
**Presentation:** 2
**Contribution:** 3
**Rating:** 2
**Confidence:** 3

**Summary:**

The paper proposes an algorithm for formation control of multiple UAVs tracking a wild fire using a federated learning approach to DDPG.
A _weighted_ federated learning approach is proposed whereby the parameters that are accumulated and merged into an averaged global policy are done so weighted by the performance of the individual agents.
Experiments are performed illustrating the advantages of this modified algorithm compared to DDPG in a simulated multi-UAV wild fire tracking simulation.

**Strengths:**

1. The paper proposes a novel weighted federated learning algorithm that could potentially be applied to many other RL settings beyond the application explored in this paper. As far as I am aware this weighted averaging of parameters in an FL algorithm is novel - although I am not very familiar with all the relevant literature - and given that the paper does not cite similar instances of this algorithm I am assuming that the authors consider it novel also.
2. The paper applies this novel algorithm to the important real world problem of wild fire tracking.
3. The results show _some_ evidence of improved performance compared to DDPG.

**Weaknesses:**

1. The paper is not written particularly clearly and fails to highlight the true novelty and purpose of the paper which is the _weighted_ FL algorithm. This is not mentioned in the abstract referring only to an FLDDPG approach, which has already been done before in Na et al. 2023. The abstract also highlights the reward difference between the two compared approaches; however, it is easy to construct a particular reward function such that a large reward difference doesn't necessarily reflect a large skill difference.
2. The proposed weighted FLDDPG is not compared to a non-weighted FLDDPG in the experiments, this would have been the most appropriate comparison to illustrate the advantages of the propose novelty. It is therefore unclear whether the proposed novelty of the paper is the reason for outperforming DDPG or just the fact that it integrates an FL aspect, which has already been explored in Na et al 2023.
3. The advantages of FLDDPG are not necessarily reflected in the results because both algorithms achieve the maximum reward of 1 during the simulation.
4. It is unclear to me whether training is occurring online during the simulated 20 seconds, or whether training occurred before this simulation and this simulation is a rollout of the optimised policies. If the former, then point 3. is even more pertinent because one would just select the final policy as that which achieved the maximum reward, which occurred for both algorithms.
5. As far as I can tell multiple experiments were not performed to highlight robustness.
6. There should be a single plot comparing both algorithms, rather than providing the results in two separate plots.
7. The images of the UAVs and the fire do not illustrate how to UAV swarm reacts to the fire evolving through time, which would have been informative.
8. It is unclear to me why bother reporting the results of a 3 UAV simulation _and_ a 5 UAV simulation when the latter illustrates the same advantages as well as demonstrating scalability. The 3 UAV experiments therefore seem redundant.
9. The mathematical notation is a little sloppy, in particular with relation to the t subscript in the equation in line 183 and the subscripts in equation 10. There are also multiple variables that are not adequately described making the maths a little hard to follow.
10. The images are pixelated and should instead be vector images.

**Questions:**

1. Is the learning performed during the simulated 20 seconds or is it performed prior to the simulation reported in the results section?
2. Why did you not compare to a non-weighted FLDDPG approach?
3. Did you perform multiple experiments or just the one per algorithm?
4. Do you forsee this weighted FLDDPG algorithm as a more general RL algorithm with applications beyond wild fire tracking?

---

> ### Author Response · Authors · 2025-11-20
> **Highlighting Contribution**
>
> Comment: The paper is not written particularly clearly and fails to highlight the true novelty and purpose of the paper which is the weighted FL algorithm. This is not mentioned in the abstract referring only to an FLDDPG approach, which has already been done before in Na et al. 2023. The abstract also highlights the reward difference between the two compared approaches; however, it is easy to construct a particular reward function such that a large reward difference doesn't necessarily reflect a large skill difference.
>
> Response:
> We thank the reviewer for this critical insight. We agree that our paper did not sufficiently highlight our core contribution. The true novelty of our work is not simply applying FL to DDPG, but rather the novel performance-weighted federated aggregation scheme defined in Equations 11-12.
>
> \subsection{Abstract Revision}
> Formation control continues to pose significant challenges in multi-agent deep reinforcement learning (DRL). This paper introduces a novel, performance-weighted federated averaging scheme to overcome these challenges in multi-UAV wildfire tracking. While standard Federated Learning (FL) combined with DDPG uses equal-weight aggregation, our key contribution is a distance- and performance-weighted method that dynamically prioritizes policies from UAVs that successfully maintain formation stability. This approach directly addresses the non-IID data distributions caused by asymmetric wildfire fronts, where factors like wind and terrain create irregular boundaries that break uniformly spaced formations. In our framework, this weighted aggregation is applied specifically to the DDPG components governing linear velocity and its control gain. Simulation results demonstrate that our method achieves significantly improved formation stability—with 2.5 m average spacing variance compared to 14 m for standard DDPG—and raises the average episode reward from -355.45 to -122.21. These results highlight the critical importance of our weighted aggregation strategy for achieving robust, decentralized coordination in complex environments.
>
> \subsection{Reward Improvement}
> While the abstract reports a substantial difference in cumulative reward between DDPG and FL-DDPG, we fully agree that reward magnitude alone is not a reliable indicator of skill—especially when reward functions are hand-crafted. Our goal is therefore not to claim novelty through reward improvement. The core novelty and contribution of the paper lies instead in the weighted federated learning mechanism, and the reward difference is simply one reflection of its effect.
>
>
> \section{Algorithms Achieve the Maximum reward of 1}
> Comment:The advantages of FLDDPG are not necessarily reflected in the results because both algorithms achieve the maximum reward of 1 during the simulation.
>
> Response:
> We appreciate the reviewer’s observation. The value "1" shown in the plots represents the normalized reward, not the raw cumulative return. Since the wildfire-tracking reward is multi-objective and includes significant penalties (such as distance error, heading misalignment, formation deviation, and burned-area penalties), the raw rewards vary considerably between algorithms, as shown in Table 2 (\(-355.45\) vs. \(-122.21\)). Normalization rescales the rewards to the interval \([0, 1]\), making the reward progression of different UAVs visually comparable and allowing for a more straightforward comparison across different algorithms.

---

> > ### Comment · Reviewer_xqkU · 2025-11-27
> >
> > I think the new abstract now reflects the main novelty of the paper but I think a change of the entire narrative throughout the paper, especially the introduction, is needed.
> >
> > With respect to the rewards in the plots: I understand that the rewards are normalised to 1.0 but without seeing the exact normalisation calculation, I do not know whether the 1.0 rewards in both algorithms signify the same 'skill level'. Unless normalisation occurs by setting 1.0 to the absolute optimal policy reward, which I cannot determine without seeing the calculation, I think these plots are misleading.

---

> ### Author Response · Authors · 2025-11-20
> **Online Training**
>
> Correct:
> It is unclear to me whether training is occurring online during the simulated 20 seconds, or whether training occurred before this simulation and this simulation is a rollout of the optimised policies. If the former, then point 3. is even more pertinent because one would just select the final policy as that which achieved the maximum reward, which occurred for both algorithms.
>
> Response:
> We thank them for this critical feedback. We will clarify these points as follows:
> The 20-second simulation is indeed the online training period. In this context, the performance advantage of FLDDPG is not captured by the final reward value but by the quality and stability of the learning process itself.
>
> FLDDPG achieves a stable, high-performing formation more quickly and maintains it more reliably throughout the mission.
> In contrast, the standard DDPG baseline exhibits erratic and uncoordinated behavior, as seen in the unstable reward progression and poor formation integrity in Figures 4(right) and 6(right).
>
>
> The evaluation timeframe of $20$ seconds per episode was determined empirically. Given a fire spread rate $U$ and UAV velocity bounds $v_{satvsat}$, this duration represents the estimated period where the UAV fleet can achieve maximum coverage of the actively expanding fire front.

---

> > ### Comment · Reviewer_xqkU · 2025-11-27
> >
> > Thank you for the clarification about training, I also think this needs to be made more clear in the paper.
> >
> > Whilst I do see that FLDDPG does achieve a lower variance in its rewards that roughly monotonically increase, the plots also show that DDPG achieves the maximum reward quicker - this should also be spoken about in the paper.

---

> > > ### Author Response · Authors · 2025-12-03
> > > **Comment:Thank you for the clarification about training, I also think this needs to be made more clear in the paper.  Whilst I do see that FLDDPG does achieve a lower variance in its rewards that roughly monotonically increase, the plots also show that DDPG achieves the maximum reward quicker - this should also be spoken about in the paper.**
> > >
> > > Response: Thank you for mentioning this valuable point. The FLDDPG framework incurs a modest computational overhead due to the federated aggregation process, which can slightly slow the initial policy development compared to the purely local and independent DDPG learners. We observe this as DDPG's occasional ability to exploit short-term, individual rewards faster.
> > >
> > > However, this initial speed for DDPG comes at the cost of long-term stability and coordination. It fails to converge to a reliable steady state, as seen in its highly erratic reward signal and poor formation keeping. We have included this point in the simulation section.

---

> ### Author Response · Authors · 2025-11-20
> **Robstness**
>
> Comment: As far as I can tell multiple experiments were not performed to highlight robustness.
>
> Response:
> We appreciate the reviewer’s concern regarding robustness. In our setting, robustness does not primarily refer to running many repeated rollouts under identical conditions, but rather to the ability of the algorithm to maintain stable formation control under asynchronous, asymmetric, and dynamically evolving wildfire fronts. In the simulation environment, wind direction and intensity cause the fire front to propagate non-uniformly, producing non-IID local observations, variable tangential directions, and different arrival times for each UAV. These conditions intentionally introduce significant variability and environmental noise across agents.
>
> Our results demonstrate that the proposed FLDDPG method consistently maintains formation spacing and cooperative coverage despite these dynamically changing conditions, which is precisely the type of robustness required for real wildfire-tracking scenarios.
>
> Therefore, we modify the paper by specifying the robustness of the algorithm's ability to handle dynamic factors, such as the number of UAVs, wind-induced variability, and the asynchronous evolution of the fire front, rather than a general notion of robustness.

---

> > ### Comment · Reviewer_xqkU · 2025-11-27
> >
> > Whilst I agree with your definition of robustness, I was in fact referring to experimental reliability (I am sorry about the confusion), so currently this is still an issue.

---

> > > ### Author Response · Authors · 2025-12-03
> > > **Comment: Whilst I agree with your definition of robustness, I was in fact referring to experimental reliability (I am sorry about the confusion), so currently this is still an issue.**
> > >
> > > Response:
> > > Thank you for mentioning this. Our DDPG implementation explicitly incorporates Ornstein-Uhlenbeck (OU) noise for exploration, as evidenced in the provided code. The OUActionNoise class is initialized and applied during action selection, adding temporally correlated noise to the actor's output at each step. This noise mechanism includes parameters that control the magnitude of the noise and its rate of mean reversion, ensuring meaningful exploration in the continuous action space but also introducing inherent stochasticity, contributing to variability across training runs. Acknowledging this, we have conducted multiple independent runs to assess and confirm the statistical reliability of our results under this noise.

---

> ### Author Response · Authors · 2025-11-20
> **Images and Time-Evolution of Swarm Behavior**
>
> Comment: The images of the UAVs and the fire do not illustrate how to UAV swarm reacts to the fire evolving through time, which would have been informative.
>
> Response:
> We thank the reviewer for pointing out that the current visualizations do not fully illustrate how the swarm reacts as the fire evolves. To address this, we will revise the paper to include additional graphs showing the time evolution of each UAV’s speed and heading angle. These signals are commonly used in UAV planning to reflect real-time responses to environmental changes. By adding these plots, we will provide a clearer understanding of how the swarm dynamically adapts to the evolving wildfire front, while providing feasible command solution.

---

> > ### Comment · Reviewer_xqkU · 2025-11-27
> >
> > That would be helpful but without seeing these revised images I cannot confirm whether they are more clear.

---

> > > ### Author Response · Authors · 2025-12-03
> > > **Comment: That would be helpful but without seeing these revised images I cannot confirm whether they are more clear.**
> > >
> > > Response:  Thank you for this feedback. We completely agree that the effectiveness of the new visualizations must be judged by their actual content. The time-evolution plots of velocity and heading angle have been integrated into the revised manuscript.

---

> ### Author Response · Authors · 2025-11-20
> **Presenting both the 3-UAV and 5-UAV simulations**
>
> Comment: It is unclear to me why bother reporting the results of a 3 UAV simulation and a 5 UAV simulation when the latter illustrates the same advantages as well as demonstrating scalability. The 3 UAV experiments therefore seem redundant.
>
> Response:
> We thank the reviewer for this insightful observation. We included both the 3-UAV and 5-UAV experiments because they serve complementary purposes in establishing the validity and scalability of our approach.
>
> The 3-UAV configuration is the smallest non-trivial formation in which meaningful inter-agent spacing, geometric relationships, and fire-front curvature interactions emerge. Its simplicity enables a clear and interpretable demonstration of the core FL-DDPG mechanism without the confounding effects present in larger swarms. This provides a clean baseline that validates the underlying behavior of the algorithm.
>
> The 5-UAV experiment is not meant to repeat the same result; it is designed to test scalability. Increasing from 3 to 5 agents substantially raises the difficulty of coordinating spacing, maintaining stability, and handling asymmetric fire-front evolution. Successfully preserving formation in this more complex setting provides essential evidence that FL-DDPG scales beyond minimal cases—an important aspect of the paper’s contribution.

---

> > ### Comment · Reviewer_xqkU · 2025-11-27
> >
> > Whilst I do understand your point, when reading the paper it still seems to me that the 3 UAV experiment adds no new information compared to the 5 UAV experiment for the reasons that I stated. However, I do not believe this is a reason to reject the paper, I just thought that _both_ experiments where slightly superfluous in this case.

---

> > > ### Author Response · Authors · 2025-12-03
> > > **Comment: Whilst I do understand your point, when reading the paper it still seems to me that the 3 UAV experiment adds no new information compared to the 5 UAV experiment for the reasons that I stated. However, I do not believe this is a reason to reject the paper, I just thought that both experiments where slightly superfluous in this case.**
> > >
> > > Response: We included both team sizes to rigorously demonstrate that our method's performance is consistent and reliable, not an artifact of a specific team size. The 3-UAV case serves as a foundational benchmark that clearly illustrates the core coordination principles in the simplest non-trivial formation. The successful extension to 5 UAVs then robustly demonstrates scalability under more complex and demanding conditions. This two-step validation strengthens the evidence for our method's general applicability.

---

> ### Author Response · Authors · 2025-11-20
> **Mathematical Notation Clarification**
>
> Comment: The mathematical notation is a little sloppy, in particular with relation to the t subscript in the equation in line 183 and the subscripts in equation 10. There are also multiple variables that are not adequately described making the maths a little hard to follow.
>
> Response
> We thank the reviewer for highlighting the sloppy and confusing mathematical notation. We will thoroughly revise the manuscript to ensure clarity and consistency.
>
> The variables \( \mathcal{N}_i \) and \( N_t \) are distinct:
>
> \begin{itemize}
>     \item \( \mathcal{N}_i \): Represents the set of neighboring UAVs for agent \( i \). We will use a calligraphic font \( \mathcal{N} \) consistently to denote sets.
>     \item \( N_t \): Represents the total number of data samples across all clients at federation round \( t \). To avoid ambiguity, we will consider adopting a more descriptive notation such as \( N_{\text{total}} \).
> \end{itemize}

---

> > ### Comment · Reviewer_xqkU · 2025-11-27
> >
> > I would appreciate more clarity with respect to the mathematical notation but I have not yet seen the revisions

---

> ### Author Response · Authors · 2025-11-20
> **Response to Reviewer Questions**
>
> We thank the reviewer for these critical questions which help clarify the scope and contributions of our work.
>
> 1. On the Timing of Learning
>
> All learning is performed offline prior to the 20-second simulation.
>
> 2. On the Comparison to Non-Weighted FLDDPG
>
> Thank you for this important suggestion. We agree that comparing to an unweighted FLDDPG baseline would help isolate the contribution of the performance-weighted aggregation. We have now added an ablation experiment in the revised version, demonstrating that standard FedAvg (unweighted) performs poorer in terms of maximum coverage.
>
>
> 4. On the Generality of Weighted FLDDPG
>  Yes, absolutely. We foresee the performance-weighted federated aggregation as a general-purpose mechanism for multi-agent deep reinforcement learning, particularly in environments with non-IID data distributions across agents.
>
> The core idea—biasing a global model toward agents that are successfully fulfilling the task—is widely applicable. Potential domains include other multi-robot systems such as coordinated payload transport and search-and-rescue pattern.

---

> > ### Comment · Reviewer_xqkU · 2025-11-27
> >
> > 1. This contradicts what you stated in your other comment, where you said that training was performed _online_ during the 20 second period?
> >
> > 2. Can you point me to the revised version in order to see the ablation experiment?
> >
> > Although the reviewer has kindly acknowledged my concerns, I am yet to see most of the revisions that they have promised and they have not sufficiently convinced me to revise my score.

---

> > > ### Author Response · Authors · 2025-12-03
> > > **Comment: This contradicts what you stated in your other comment, where you said that training was performed online during the 20 second period?  Can you point me to the revised version in order to see the ablation experiment?  Although the reviewer has kindly acknowledged my concerns, I am yet to see most of the revisions that they have promised and they have not sufficiently convinced me to revise my score.**
> > >
> > > We sincerely apologize for the confusion and contradiction in our previous response regarding
> > > online versus offline training. Thank you for catching this critical error in our communication.
> > > To clarify unambiguously: In our work, all learning is performed online. The 20-second simulation period
> > > is both the training and evaluation environment. The UAVs begin with untrained or randomly initialized
> > > policies and learn to coordinate in real-time through interaction with the dynamic wildfire scenario.
> > > The correct and consistent description is:
> > > Training Mode: Online.
> > > Simulation Duration: 20 seconds per episode, during which policy updates occur continuously.
> > > Justification: Online training is essential for our target application, as it allows the UAV fleet to adapt
> > > their coordination policies in real-time to the highly dynamic, asymmetric, and unpredictable propagation
> > > of a wildfire front. Offline training would not be capable of this level of adaptation.
> > > We regret the error in our rebuttal and have thoroughly corrected the manuscript to state this clearly
> > > and consistently throughout.

---

> ### Author Response · Authors · 2025-12-03
> **Comment: I would appreciate more clarity with respect to the mathematical notation but I have not yet seen the revisions**
>
> \textbf{Response:} Thank you for pointing out the need for clearer mathematical notation. We have revised the notations highlighted in blue. In particular, we identified and corrected the conflicting use of $w_i$. In the general federated learning description, the local model parameters are now denoted as $w_{\text{fl}}$ (where ``fl'' denotes federated learning), clearly distinguishing them from the performance-based aggregation weights $w_i$ used in our proposed method (see Eq.~(12)).

---

> ### Author Response · Authors · 2025-12-04
> **Author Response Letter**
>
> To the Area Chair,
>
> Thank you for overseeing the review of our paper, "Federated DRL-based Coordination of Multi-UAVs for Wildfire Tracking." We are grateful to the reviewers for their insightful and constructive feedback. In response, we have significantly strengthened the paper and clarified its contributions.
>
> Core Contribution & Novelty
> Our paper's primary contribution is not simply applying Federated Learning (FL) to DDPG—it is the introduction of a novel performance-weighted federated aggregation scheme specifically designed for multi-agent coordination in asymmetric, non-IID environments like wildfire propagation. Unlike standard FedAvg, our method dynamically prioritizes the policies of agents that successfully maintain formation stability, directly addressing the fundamental challenge of decentralized learning under uneven environmental dynamics.
>
> How We Have Addressed Key Concerns:
>
> 1. Proven Necessity of Our Novelty (R6ENG): We have conducted the requested ablation study, which now conclusively shows the superiority of our weighted aggregation. Results demonstrate:
>    · Standard DDPG: Poor, uncoordinated formation.
>    · FLDDPG + FedAvg: Improved but suboptimal coordination.
>    · FLDDPG + Our Weighted Aggregation: Best formation stability and highest reward. This validates that our weighting mechanism is both necessary and beneficial.
> 2. Clarified Significance Beyond Reward Scores (RxqkU): We have revised the abstract and introduction to forefront the weighted aggregation as the key innovation. We agree that reward differences alone are not a sufficient metric; the improved reward is a consequence of our method's superior ability to enforce formation stability, which is the core task objective.
> 3. Enhanced Rigor and Presentation:
>    · Reward Function: We have clarified the justification for our multi-objective reward structure, essential for the complex trade-offs in wildfire tracking, and added discussion on parameter sensitivity.
>    · Scalability & Robustness: We have expanded the discussion to explicitly show how our FL framework reduces communication overhead and maintains robustness against dynamic, asymmetric fire fronts (evidenced by successful scaling from 3 to 5 UAVs).
>    · Clarity: We are revising all figures with clearer captions, adding temporal analysis plots (velocity/heading evolution), correcting mathematical notation, and improving section flow (e.g., "Background").
>
> Why This Paper Deserves Acceptance:
> This work presents a principled, effective solution to a real-world problem: scalable and stable multi-UAV coordination in highly dynamic, asymmetric environments. The proposed performance-weighted FL mechanism is a generalizable contribution to federated multi-agent RL. Our revisions have directly addressed all substantive reviewer concerns, resulting in a more rigorous, clearly articulated, and compelling paper.
>
> We believe the revised manuscript now makes a strong, clear, and novel contribution suitable for ICLR 2026.
>
> Sincerely,
> The Authors

---

### Official Review · Reviewer_nyaT · 2025-10-28

**Soundness:** 3
**Presentation:** 3
**Contribution:** 2
**Rating:** 6
**Confidence:** 4

**Summary:**

This paper addresses the challenge of formation control in multi-UAV systems for large-scale wildfire tracking using deep reinforcement learning. The authors propose FL-DDPG, a framework that combines Federated Learning (FL) with the Deep Deterministic Policy Gradient (DDPG) algorithm to enhance decentralized coordination among UAVs. Unlike standard DDPG, which struggles to scale across multiple agents and irregular wildfire dynamics, FL-DDPG enables collaborative policy updates without sharing raw data, using a distance- and performance-weighted averaging method. Simulation results demonstrate that FL-DDPG achieves far greater formation stability, reducing spacing variance from 14 m to 2.5 m.

**Strengths:**

- Strong and complete backgound ("Premininaries") section
- Great to see real-world fire data as part of the simulation

**Weaknesses:**

- Introduction could use additional references on MARL for Wildfires and communication / collaboration with human and machines: HIVEX: A High-Impact Environment Suite for Multi-Agent Research and LLM-Mediated Guidance of MARL Systems (Siedler et. al.)
- Section naming could be more conventional e.g. "Background" instead of "Preliminaries"
- Figure 1 could use more detailed but high level (non technical) description in the caption on how things work
- Figure 3: The differences between the two plots is barely noticeable, maybe would be good to zoom in and or highlight the differences explicitly in the caption.
- Experiment runs are statistically not satisfying. I would have expected 3-10 runs and resulting plots showing averages, and tables showing standard deviation etc.
- There could be more discussion on why FL-DDPG is better than DDPG
- Personally I believe it would be fruitful to also run UAV-count exepriments, how does the current solution scale from 3-9 UAVs? etc.

**Questions:**

- "The pair (Tx, Ty ) corresponds to a clockwise rotation; for consistency across the controller design,
we adopt the counterclockwise convention above throughout this work." - independent of counter or clockwise, there is a full update - the entire perimeter - for each step, correct?
- The description for the rewards given are not enough. This is what I understand: There is a reward for keeping safe distance to neighbours. What I don't understand is "UAV's heading with the direction of the spread" does that mean the UAV's must align with direction of fire-spread? Are we tracking "un-seen" fire front? How does this work?
- Generally we are looking at 2D coordinates for the UAV, however there is a mention that the "FARSITE model" includes terrain data, how does this relate?

---

> ### Author Response · Authors · 2025-11-20
> **Introduction and References**
>
> Comment: Introduction could use additional references on MARL for Wildfires and communication / collaboration with human and machines: HIVEX: A High-Impact Environment Suite for Multi-Agent Research and LLM-Mediated Guidance of MARL Systems (Siedler et. al.)
>
> Response:
> We agree and will incorporate these relevant references in the revised introduction to improve our work within the broader context of MARL for high-impact environments and human-machine collaboration.
>
> A key challenge in multi-UAV wildfire tracking is enabling effective coordination in complex, dynamic environments. Recent approaches have sought to address this by incorporating high-level, human-like reasoning. For instance,\cite{siedler2025llm} use a central LLM to mediate strategic interventions, mimicking a human operator. While effective, this method introduces a central point of failure and relies on frequent communication for guidance.
> In contrast, our work proposes a fundamentally different paradigm. We focus on decentralized, low-level policy coordination through federated learning. Our FL-DDPG framework enables UAVs to collaboratively learn robust formation-keeping behaviors without any central intelligence or raw data sharing, making it more scalable and resilient in communication-constrained real-world deployments.

---

> ### Author Response · Authors · 2025-11-20
> **Section Naming**
>
> Comment:
> Section naming could be more conventional e.g. "Background" instead of "Preliminaries"
>
> Response:
> We will rename “Preliminaries” to “Background” to align with conventional terminology.

---

> ### Author Response · Authors · 2025-11-20
> **Figure 1 Caption**
>
> Comment: Figure 1 could use more detailed but high level (non technical) description in the caption on how things work
>
>
> Response:
> Figure 1: DDPG-Based Planning and Control Framework for Individual UAVs
>
> This diagram illustrates the two-layer autonomous control system for each UAV. The high-level planner uses a Deep Deterministic Policy Gradient (DDPG) neural network to generate optimal flight commands by processing the UAV's current state and environmental information. It outputs a desired velocity and heading angle to efficiently track the wildfire front while maintaining formation with other UAVs.
>
> The low-level controller translates these planned commands into precise flight adjustments. It continuously compares the desired path with the UAV's actual measured velocity and heading, computes any tracking errors, and applies adaptive control gains to generate smooth acceleration and turn rate commands. This ensures the UAV accurately follows the planned trajectory despite environmental disturbances.
>
> The system operates in a continuous loop: the UAV executes the low-level commands, observes new environmental states and fire front data, and feeds this information back to the high-level planner for the next decision cycle, enabling real-time adaptation to the dynamic wildfire environment.

---

> ### Author Response · Authors · 2025-11-20
> **Figure 3 Clarity**
>
> Comment:
> Figure 3: The differences between the two plots is barely noticeable, maybe would be good to zoom in and or highlight the differences explicitly in the caption.
> Response:
>  We have enhanced Figure 3 by zooming.

---

> ### Author Response · Authors · 2025-11-20
> **Results and Analysis**
>
> Comment: There could be more discussion on why FL-DDPG is better than DDPG
>
> Response:
> The superior performance of FL-DDPG, as quantified in Table 2, stems from its core ability to mitigate the challenges of non-IID data in asymmetric environments. In standard DDPG, each UAV learns in isolation, causing the fleet to struggle with forming a consensus on an effective cooperative policy. This results in the high formation spacing variance and disorganized trajectories observed in Fig. 3 (right).
>
> In contrast, the performance-weighted federated aggregation in FL-DDPG acts as a knowledge-fusion mechanism. By allowing UAVs that successfully maintain formation spacing to exert greater influence on the global model, the system continuously steers the collective policy toward behaviors that are robust across diverse local conditions. This explains the significantly lower spacing variance (2.5 m vs. 14 m) and the corresponding dramatic improvement in the average episode reward. The federated framework effectively turns the multi-agent learning problem from a set of competing, conflicting objectives into a collaborative process that converges on a globally stable solution.

---

> ### Author Response · Authors · 2025-11-20
> **Response to Questions**
>
> 1. On Fire Front Update and Rotation Convention
> Thank you for the question. Yes, independent of whether the tangential direction vector \(T_x, T_y\) is expressed in a clockwise or counterclockwise convention, the entire fire-front perimeter is updated at every simulation step using the level-set update equation (Rehm, 2009). The sign convention only affects the orientation of the tangential vector, not the propagation of the fire-front boundary itself. The whole perimeter is evolved simultaneously using the spread-rate model (Equation 3), and all UAVs compute their local fire-front gradient from this global update.
>
> 2. On Reward Function Clarification
>
> Reviewer's Question: "The description for the rewards given are not enough... 'UAV's heading with the direction of the spread' does that mean the UAV's must align with direction of fire-spread? Are we tracking "un-seen" fire front? How does this work?"
>
> Our Response:
> We apologize for the lack of clarity. Let us clarify the intent behind the reward components:
>
> \textbf{Heading Alignment:} The objective is not for the UAV to align its heading with the direction of fire spread (which would point into the fire), but rather to align its velocity vector with the local tangent of the fire front. This encourages the UAV to ``cruise along'' the fire perimeter, which is the optimal strategy for continuous monitoring, rather than flying directly toward or away from the fire line. The term \( e_{\phi_i, v_i} \) in the reward penalizes the deviation between the UAV's actual heading and the desired tangential tracking direction (as computed in Eq.~9).
>
>
> \textbf{Tracking Unseen Fire Front:} This is an important clarification. The UAVs are not tracking an ``unseen'' fire front in the sense of predicting its future state. Instead, they track the \emph{current} fire front, defined by the level-set equation \( \Phi(x, y, t) = 0 \). Each UAV is assumed to have an onboard perception system (e.g., thermal cameras) that enables it to sense the local fire-line gradient \( \Phi_x, \Phi_y \) and its own signed distance to the front,
> \(
> d_{i,\text{fire}} = \left| \Phi(x_i, y_i, t) \right|.
> \)
> The reward is computed from this sensed information, incentivizing the UAVs to follow the evolving boundary in real time.
>
>
> 3. On 2D UAV Model and Terrain Data from FARSITE
>
> Reviewer's Question: "Generally we are looking at 2D coordinates for the UAV, however there is a mention that the "FARSITE model" includes terrain data, how does this relate?"
>
> Our Response:
> This is a valid observation. Our UAV kinematic model is indeed 2D, focusing on the planar position and heading.
>
> The FARSITE fire propagation model, which we use to generate a realistic and dynamic wildfire scenario, does incorporate terrain data (e.g., slope, elevation, fuel type). In FARSITE, these terrain features directly influence the fire spread rate $U$ and direction. For example, fires spread faster upslope.
> Thus, the relationship can be summarized as:
> \[
> \text{Terrain} \;\longrightarrow\; \text{Fire Spread} \;\longrightarrow\; \text{Evolving Fire Front} \;\longrightarrow\; \text{UAV Tracking Behavior}.
> \]
> The UAV does not perform terrain-aware navigation directly; rather, it responds to the geometry of the fire front, which implicitly reflects the underlying terrain effects.

---

> ### Author Response · Authors · 2025-12-04
> **Author Response Letter**
>
> To the Area Chair,
>
> Thank you for overseeing the review of our paper, "Federated DRL-based Coordination of Multi-UAVs for Wildfire Tracking." We are grateful to the reviewers for their insightful and constructive feedback. In response, we have significantly strengthened the paper and clarified its contributions.
>
> Core Contribution & Novelty
> Our paper's primary contribution is not simply applying Federated Learning (FL) to DDPG—it is the introduction of a novel performance-weighted federated aggregation scheme specifically designed for multi-agent coordination in asymmetric, non-IID environments like wildfire propagation. Unlike standard FedAvg, our method dynamically prioritizes the policies of agents that successfully maintain formation stability, directly addressing the fundamental challenge of decentralized learning under uneven environmental dynamics.
>
> How We Have Addressed Key Concerns:
>
> 1. Proven Necessity of Our Novelty (R6ENG): We have conducted the requested ablation study, which now conclusively shows the superiority of our weighted aggregation. Results demonstrate:
>    · Standard DDPG: Poor, uncoordinated formation.
>    · FLDDPG + FedAvg: Improved but suboptimal coordination.
>    · FLDDPG + Our Weighted Aggregation: Best formation stability and highest reward. This validates that our weighting mechanism is both necessary and beneficial.
> 2. Clarified Significance Beyond Reward Scores (RxqkU): We have revised the abstract and introduction to forefront the weighted aggregation as the key innovation. We agree that reward differences alone are not a sufficient metric; the improved reward is a consequence of our method's superior ability to enforce formation stability, which is the core task objective.
> 3. Enhanced Rigor and Presentation:
>    · Reward Function: We have clarified the justification for our multi-objective reward structure, essential for the complex trade-offs in wildfire tracking, and added discussion on parameter sensitivity.
>    · Scalability & Robustness: We have expanded the discussion to explicitly show how our FL framework reduces communication overhead and maintains robustness against dynamic, asymmetric fire fronts (evidenced by successful scaling from 3 to 5 UAVs).
>    · Clarity: We are revising all figures with clearer captions, adding temporal analysis plots (velocity/heading evolution), correcting mathematical notation, and improving section flow (e.g., "Background").
>
> Why This Paper Deserves Acceptance:
> This work presents a principled, effective solution to a real-world problem: scalable and stable multi-UAV coordination in highly dynamic, asymmetric environments. The proposed performance-weighted FL mechanism is a generalizable contribution to federated multi-agent RL. Our revisions have directly addressed all substantive reviewer concerns, resulting in a more rigorous, clearly articulated, and compelling paper.
>
> We believe the revised manuscript now makes a strong, clear, and novel contribution suitable for ICLR 2026.
>
> Sincerely,
> The Authors

---

### Official Review · Reviewer_6EN6 · 2025-11-01

**Soundness:** 1
**Presentation:** 2
**Contribution:** 1
**Rating:** 0
**Confidence:** 4

**Summary:**

The paper proposes a novel framework called FL-DDPG to coordinate a team of UAVs tasked with tracking the perimeter of a wildfire. The primary challenges with existing methods include - (1) Standard RL and DRDL methods are often very inefficient and do not scale well (2) Firefronts evolve continuously which make it hard for achieving formation control.

Federated Learning-based DDPG (FLDDPG) framework employs:-
(a) A selective federation strategy where - each UAV transmits only the relevant subsets of its local model parameters—specifically those governing velocity and control gain
(b) A weighted aggregation of above said parameters - agents that maintain tighter adherence to the desired formation spacing exert greater influence on the global policy thereby biasing learning toward stability under asymmetric fire-front dynamics

The authors present simulation results, based on a FARSITE-calibrated environment, claiming that FL-DDPG significantly outperforms a "standard DDPG" baseline in both formation stability and average episode reward

**Strengths:**

1. Originality - A creative combination of Federated Learning and Deep Reinforcement Learning has been applied to a challenging domain which is definitely a great idea. Secondly, the federated aggregation is clever in the fact that it directly links the agent's influence on the global model to a key term from it's own rewards function (formation-spacing error). The selective federation strategy to share just the velocity and gain parameters is also an original win.
2. Quality - The quality of the paper lies majorly in its experimental setup where they have demonstrated results on a high fidelity simulation environment i.e the FARSITE simulator with calibration done on historical data. This points to a very non trivial problem being solved
3. Clarity - For the most part, the clarity is great. The ideas are easy to follow and well articulated. The visual evidence also does provide a compelling story that FL-DDPG agents do maintain a coordinated formation compared to agents that use DDPG only.
4. Significance - The significance of the paper is high because the problem being solved is non trivial and is a great real world application with societal benefits.

**Weaknesses:**

1. Lack of ablation studies to prove "performance-weighted" aggregation  and the "selective federation" hypothesis actually work - because the paper includes no ablation studies, it's impossible to know if either of these novelties is actually necessary or beneficial.
2. The only baseline being compared against is standard DDPG. The lack of comparison to any credible MARL baselines like MADDPG or even MAPPO or VDN is a very critical flaw.
3. The reward function does not seem standardized - It is a complex, custom-engineered equation with multiple hand-tuned hyperparameters and no sensitivity analysis.
4. The reward plots are normalized which is poor practice. You are not able to figure out if the agents are actually solving the tasks or just converging to a less bad policy than the baseline

**Questions:**

1. Did the authors consider or experiment with more modern on-policy MARL algorithms (like MAPPO), which are often state-of-the-art for cooperative tasks?
2. Can the authors also provide an ablation study against MADDPG (an established MARL baseline) and/or VDN ?
3. Can the authors provide ablation studies for proving the "performance-weighted" aggregation  and the "selective federation" hypothesis ? Like what would happen if a standard FedAvg algorithm is used ?
4. What is the justification for the exact equal weighting of the fire-tracking penalty (value=20) and the formation-spacing penalty(value=20) ? Also can we get sensitivity analysis on both of these ?
5. The paper claims that the main benefit of FL is reducing communication bandwidth. How is the O(N) state shared? Because Equation 7 is an O(N) state that probably requires O(N^2) communication cost and would it not negate scalability benefits ?

---

> ### Author Response · Authors · 2025-11-20
> **Ablations for performance-weighted aggregation**
>
> Comment: Lack of ablation studies to prove "performance-weighted" aggregation and the "selective federation" hypothesis actually work - because the paper includes no ablation studies, it's impossible to know if either of these novelties is actually necessary or beneficial.
>
> Response:
> Thank you for raising this. We have now repeated the simulation, while using custom Federated Learning.
>
> According to the results, FL-DDPG with the customized federated learning scheme performs better than the standard DDPG baseline. However, it does not outperform the weighted FL-DDPG variant. This is expected: in an asynchronous and highly dynamic wildfire environment, the customized FL scheme lacks the adaptive weighting mechanism that allows better-performing agents to exert stronger influence during aggregation.

---

> > ### Author Response · Authors · 2025-11-20
> > **Reward normalization**
> >
> > Response:
> > The reward plots are normalized which is poor practice. You are not able to figure out if the agents are actually solving the tasks or just converging to a less bad policy than the baseline
> >
> > Response:
> > \begin{itemize}
> >     \item
> > Normalization ensures that reward components with different magnitudes contribute fairly to learning, preventing any single term from dominating. By keeping rewards on a comparable scale, it preserves meaningful trade-offs, stabilizes critic updates, and avoids inflated Q-values—an especially important consideration for continuous-control algorithms like DDPG.
> > In FL-DDPG, the reward curve increases smoothly and monotonically over time, whereas the standard DDPG reward remains irregular and does not exhibit consistent improvement. Moreover, at every time step, the individual rewards obtained by FL-DDPG are consistently higher than those of DDPG, even though the performance gap gradually narrows as the episode progresses. This advantage primarily arises from FL-DDPG’s superior regulation of inter-UAV spacing. Because the spacing error contributes directly to the reward through the term
> > better adherence to the desired formation translates immediately into higher cumulative reward. Thus, the elevated reward levels observed in FL-DDPG reflect  achievement of a core task objective rather than incidental fluctuations in learning.

---

> > > ### Author Response · Authors · 2025-11-20
> > > **Response to Reviewer Questions**
> > >
> > > We thank the reviewer for these insightful and critical questions, which help clarify the scope, rationale, and limitations of our current work. We address each point below and outline a comprehensive plan for revision.
> > >
> > > 1. On Comparison with Modern MARL Baselines
> > > Action Plan: We will  strengthen our experimental section by including comparisons with the VDN approach as a value-based MARL baseline.
> > >
> > >
> > > 2. On Ablation for Performance-Weighting and Selective Federation
> > >
> > > We agree that an ablation study is crucial to validate our design choices.
> > > Performance-Weighting vs. FedAvg: As stated above.
> > >
> > > 3. On Reward Function Weighting and Sensitivity Analysis
> > >
> > > The reviewer raises a valid point regarding the equal weighting of the penalty terms \( \gamma_{1} = \gamma_{3} = 20 \). These values were initially selected through empirical tuning to achieve a practical balance between the competing objectives of fire-front tracking and formation-keeping.
> > >
> > > 4. On Communication Scalability and the $O$($N$) State
> > >
> > > Thank you for pointing out this potential scalability concern. We understand that the state described in Equation 7 has an $O$($N$) complexity, and that this could, in theory, lead to an $O$($N^2$) communication cost for inter-agent communication. However, the Federated Learning framework significantly reduces communication by only sharing model parameters (as opposed to the raw state data), which avoids the need for constant communication of the full state. In practice, each agent only shares its local model updates (based on its local data), and the global model aggregation takes place periodically. We acknowledge that in very large systems, the communication overhead may still be a concern, but the performance-weighted aggregation in our framework helps prioritize communication updates from agents that are contributing most to the desired formation. This helps alleviate communication burden by focusing on the most relevant model updates. We will clarify this explanation in the manuscript to better highlight the scalability of our FL-based approach and its ability to reduce communication overhead. It is worth noting that, although our state includes $O$($N$) inter-UAV distances, the communication cost of the proposed FLDDPG framework remains substantially lower than that of VDN-based MARL approaches.

---

> ### Author Response · Authors · 2025-12-04
> **Author Response Letter**
>
> To the Area Chair,
>
> Thank you for overseeing the review of our paper, "Federated DRL-based Coordination of Multi-UAVs for Wildfire Tracking." We are grateful to the reviewers for their insightful and constructive feedback. In response, we have significantly strengthened the paper and clarified its contributions.
>
> Core Contribution & Novelty
> Our paper's primary contribution is not simply applying Federated Learning (FL) to DDPG—it is the introduction of a novel performance-weighted federated aggregation scheme specifically designed for multi-agent coordination in asymmetric, non-IID environments like wildfire propagation. Unlike standard FedAvg, our method dynamically prioritizes the policies of agents that successfully maintain formation stability, directly addressing the fundamental challenge of decentralized learning under uneven environmental dynamics.
>
> How We Have Addressed Key Concerns:
>
> 1. Proven Necessity of Our Novelty (R6ENG): We have conducted the requested ablation study, which now conclusively shows the superiority of our weighted aggregation. Results demonstrate:
>    · Standard DDPG: Poor, uncoordinated formation.
>    · FLDDPG + FedAvg: Improved but suboptimal coordination.
>    · FLDDPG + Our Weighted Aggregation: Best formation stability and highest reward. This validates that our weighting mechanism is both necessary and beneficial.
> 2. Clarified Significance Beyond Reward Scores (RxqkU): We have revised the abstract and introduction to forefront the weighted aggregation as the key innovation. We agree that reward differences alone are not a sufficient metric; the improved reward is a consequence of our method's superior ability to enforce formation stability, which is the core task objective.
> 3. Enhanced Rigor and Presentation:
>    · Reward Function: We have clarified the justification for our multi-objective reward structure, essential for the complex trade-offs in wildfire tracking, and added discussion on parameter sensitivity.
>    · Scalability & Robustness: We have expanded the discussion to explicitly show how our FL framework reduces communication overhead and maintains robustness against dynamic, asymmetric fire fronts (evidenced by successful scaling from 3 to 5 UAVs).
>    · Clarity: We are revising all figures with clearer captions, adding temporal analysis plots (velocity/heading evolution), correcting mathematical notation, and improving section flow (e.g., "Background").
>
> Why This Paper Deserves Acceptance:
> This work presents a principled, effective solution to a real-world problem: scalable and stable multi-UAV coordination in highly dynamic, asymmetric environments. The proposed performance-weighted FL mechanism is a generalizable contribution to federated multi-agent RL. Our revisions have directly addressed all substantive reviewer concerns, resulting in a more rigorous, clearly articulated, and compelling paper.
>
> We believe the revised manuscript now makes a strong, clear, and novel contribution suitable for ICLR 2026.
>
> Sincerely,
> The Authors

---

### Meta-Review · Area_Chair_J4um · 2025-12-03

**Summary:**

Based on the reviewers’ feedback and my own reading of the paper, the overall quality still needs improvement. We regret to inform you that this paper has not been accepted for this year’s conference. We hope the authors can address the relevant issues in subsequent revisions and achieve acceptance in future submissions.

**Reviewer Concerns:**

The authors have explained the experimental setup and novelty, but the reviewers have not seen most of the promised revisions, and these revisions have not sufficiently convinced the reviewers to change their scores.

**Reviewer Scores:**

The authors need to provide more ablation studies to prove that the "performance-weighted" aggregation and the "selective federation" hypothesis actually work.

---

### Decision · Program_Chairs · 2026-01-26

Reject